# The Impact of Climate Change on Extreme Winds over Northern Europe According to CMIP6

Xiaoli Guo Larsén<sup>1</sup>, Marc Imberger<sup>1</sup>, Ásta Hannesdóttir<sup>1</sup>, and Andrea N. Hahmann<sup>1</sup> <sup>1</sup>Department of Wind and Energy Systems, Technical University of Denmark **Correspondence:** Xiaoli Guo Larsén (xgal@dtu.dk)

**Abstract.** We study the possible effect of climate change on the extreme wind over Northern Europe using data from 18 models of the Sixth Phase of the Coupled Model Intercomparison Project (CMIP6) and the high-emission SSP585 scenario. We use the spectral correction method to correct the 6-hourly wind speeds and calculate the 50-year wind at an equivalent temporal resolution of 10 minutes, consistent with the International Electrotechnical Commission (IEC) standard. We assess the quality

- 5 of the CMIP6 wind data during the historical period through comparison to the spatial patterns of the extreme wind in three reanalysis data. We obtain the possible effect of climate change through the comparison of the extreme wind parameters, including the 50-year wind and the 95%-percentile of the wind speed, and the change in turbine class at 50 m, 100 m and 200 m, between a near future period (2020–2049) and the historic period (1980–2009). The analysis shows an overall increase in the extreme winds in the North Sea and the southern Baltic Sea, but a decrease over the Scandinavian Peninsula and most of
- 10 the Baltic Sea. However, the analysis is inconclusive to whether higher or lower classes of turbines will be installed in this area in the future.

# 1 Introduction

Extreme winds can cause great loss to society and the 50-year wind,  $U_{50}$ , a common measure of extreme winds, is one 15 of the most important siting parameters that needs to be estimated when planning regional wind energy development. An accurate estimation of the extreme wind can help to harvest more electricity from winds, while avoiding placing the turbines in dangerous places, and to avoid or over-design of the wind turbines; it is directly related to the Levelized Cost Of Energy (LCOE) and the cost of climate mitigation.

We live in a world where climate is changing and where the damage to society from extreme winds is rising every year (MunichRe, 2011). The scale and speed of wind energy development and deployment have never been so large and it will continue in the future (GWEC, 2022). It is thus a relevant question to ask how climate change (CC) will impact extreme winds in the future and what this change implies for the cost of wind energy development.

While observed trends and the effect of CC has been studied extensively for extreme temperature and precipitation (e.g. IPCC, 2021, Chapter 11), studies on extreme winds are fewer. For the studies of extreme wind, data from various models with a variety of temporal and spatial resolution have been used. Reanalysis data has been a popular choice for extreme wind 25 estimation due to its global coverage and long-term availability, including ECMWF reanalysis ERA-40 (e.g. Della-Marta et al., 2009), reanalysis from National Centers for Environmental Prediction and the National Center for Atmospheric Research (NCEP/NCAR) (e.g. Larsén and Mann, 2009), NOAA 20th century reanalysis (e.g. Donat et al., 2011), the 5th generation reanalysis from ECMWF ERA5 (e.g. Pryor and Bartelmie, 2021; Imberger and Larsén, 2022), Climate Four-Dimensional Data Assimilation CFDDA (e.g. Hansen et al., 2016), Climate Forecast System Reanalysis I CFSR (e.g. Larsén and Kruger, 30 2014) and Modern-Era Retrospective analysis for Research and Applications MERRA2 (e.g. Imberger and Larsén, 2022). The temporal resolution of these data ranges from 1 to 6 hours, and the spatial grid spacing from about 25 km to a couple of hundreds of kilometers. This resolution is too coarse and not ideal for site-specific extreme wind calculations. Using data from mesoscale model simulations is expected to improve the extreme wind estimation, as shown in e.g. Bastine et al. (2018) and 35 Larsén et al. (2021), where data from the Weather Research and Forecasting (WRF) model with spatial grid spacing of 3 km were used. The only global coverage of extreme wind that has applied microscale modeling with spatial grid spacing of 275 m

For extreme winds, many of the studies focus on the estimation of extreme winds using historical data, rather than about the impact of CC. In recent decades, Regional Climate Models (RCMs) have been used to study the impact of CC on extreme

has been created within the in the Global Atlas for Siting Parameters (GASP) project (Larsén et al., 2022).

- wind. These include, for example, the PRUDENCE project (Prediction of Regional Scenarios and Uncertainties for Defining European Climate Change Risks and Effects), a EU Framework Program 6 project that ended in 2004 (PRU), ENSEMBLES (Ensemble-Based Predictions of Climate Changes and their Impacts), a successor to PRUDENCE which finished in 2009 (ENS) and CORDEX (Coordinated Regional Downscaling Experiment, Giorgi and Gutowski, 2015), based on infrastructure known from the Fifth Phase of the Coupled Model Intercomparison Project (CMIP5; Taylor et al., 2012). The RCMs included in the
- PRUDENCE project have a spatial grid spacing of 50 km, ENSEMBLES 25 km and CORDEX 12 km. With PRUDENCE, Beniston et al. (2007) (with both high and low emission scenarios A2 and B2), Schwierz et al. (2010) (with SRES A2 scenario), and Rockel and Woth (2007) (also with SRES A2 scenario) studied the CC on extreme wind using 90th, 98th and 99th percentiles of wind speed, as well as gust and the *T*-year return wind speed for the period 1961–1990 vs. 2071–2100. They found that these models suggest an increase in extreme wind in Central Europe in the future due to climate change. With
- ENSEMBLE, Donat et al. (2011) (with SRES A1B scenario) analyzed the 98th percentile of wind speed and found an increase in Northern Europe and a decrease in the Mediterranean Sea; Pryor et al. (2012) (with SRES A1B scenario) analyzed  $U_{50}$  and found similar change in Northern Europe in line with findings by Donat et al.; Clausen et al. (2012) (with SRES A1B scenario) analyzed four ensemble model members and calculated the 90th, 95th and 99th percentiles,  $U_{50}$  and the linear trend at hub height of 100 m, and found that in Europe, there is no significant change in  $U_{50}$  in majority of the grid cells by the middle of
- the century, while towards the end of the century, there is an increasing number of grid cells that show increases in  $U_{50}$  larger than the natural variability. With CORDEX, Outten and Sobolowski (2021) (with RCP8.5 scenario) calculated 10, 20, 30, 50

and 100-year return wind values for Europe for three periods 2011–2040, 2041–2070 and 2071–2100 and they found an overall increase in extreme wind in the future, for both Northern and Southern Europe.

- In comparison with estimations from ENSEMBLE data, Outten and Sobolowski (2021) found that the 12 km resolution 60 CORDEX data provides more details over complicated landscape e.g. land and coastal areas, with enhanced extreme events. The typical resolution of RCMs, which is on the order of tens of kilometers on hourly basis, is not high enough for the purpose of site specific estimation of extreme wind, due to the ever-present smoothing effect in numerical models (Larsén et al., 2012). These data are however still useful to identify trends and changes in the extreme wind estimation.
- The CMIP6 project is the sixth phase and most recent phase of the CMIPs (Eyring et al., 2016). It used World Climate Research Programme (WCRP) Grand science Challenges (GCs) as the scientific backdrop of its experiment design. These GCs constitute a main component of the WCRP strategy to accelerate progress in climate science (Brasseur and Carlson, 2015), with one of the seven subjects particularly on climate extremes: "assessing climate extremes, what controls them, how they have changed in the past and how they might change in the future". A parallel study, Hahmann et al. (2022), used this data set to examine the CC effect on wind resources for Northern Europe.
- Previous studies on the impact of CC on the extreme wind using the several generations of RCMs have not reached a consensus. In this study, we make use of the CMIP6 data to study the CC impact on the extreme wind. To relate our calculation and analysis to the IEC standard for turbine design (IEC, 2019), we downscale the CMIP6 time series of wind speed in the temporal domain using a fast and simple spectral model. This method, called the spectral correction (SC) method, produces an equivalent 10-min temporal resolution and is introduced in section 2.3. Data are introduced in section 2, followed by the
- results in section 3. The discussion of the results and the conclusions are in section 4 and section 5, respectively.

## 2 Data and methods

We use the output from CMIP6 models to assess the CC impact on the extreme wind over two periods: the historical period (his-Period, 1980–2009) and the near future period (fut-Period, 2020–2049) from the CMIP6 historical and SSP585 simulations, respectively (Eyring et al., 2016). These two periods are relevant for a turbine's life time of 20–30 years for a wind farm being

planned in 2020. When using climate model output, it is always a challenge to quantify or qualify the systematic signals and uncertainties. To help qualifying the model reliability, we compare the estimate of  $U_{50}$  from CMIP6 his-Period to that from the reanalysis data. The comparison results will be taken into consideration when we perform the analysis of CC.

The CMIP6 data are introduced in section 2.1 and the reanalysis data in section 2.2. Both data will be processed using the SC method when calculating  $U_{50}$ , which is described in section 2.3.

# 85 2.1 The CMIP6 data

To be consistent with the study of wind resource using CMIP6 data in Hahmann et al. (2022), we use the 18 models listed in Table 1. Among the many CMIP6 models these 18 are chosen because of the availability of model outputs of surface pressure, temperature and humidity in addition to the wind components u and v at the raw model level for the simulations of the historical

Table 1. Models in the CMIP6 archive used in this study.

| No. | Model           | Approx. grid                       | Number of vertical | Reference                 |
|-----|-----------------|------------------------------------|--------------------|---------------------------|
|     |                 | spacing                            | levels             |                           |
| 1   | ACCESS-CM2      | $1.25^\circ\times 1.875^\circ$     | 85                 | Tilo et al. (2020)        |
| 2   | CanESM5         | $2.8125^\circ \times 2.79^\circ$   | 49                 | Swart et al. (2019)       |
| 3   | CESM2           | $1.25^\circ \times 0.94^\circ$     | 32                 | Danabasoglu et al. (2020) |
| 4   | CMCC-CM2-SR5    | $1.25^\circ \times 0.94^\circ$     | 30                 | Cherchi et al. (2019)     |
| 5   | CNRM-CM6-1      | $1.4^{\circ} \times 1.4^{\circ}$   | 91                 | Voldoire et al. (2019)    |
| 6   | CNRM-ESM2-1     | $1.4^{\circ} \times 1.4^{\circ}$   | 91                 | Séférian et al. (2019)    |
| 7   | HadGEM3-GC31-LL | $1.875^\circ\times1.25^\circ$      | 85                 | Sellar et al. (2020)      |
| 8   | HadGEM3-GC31-MM | $0.833^\circ \times 0.556^\circ$   | 85                 | Sellar et al. (2020)      |
| 9   | IPSL-CM6A-LR    | $2.5^\circ\times 1.27^\circ$       | 91                 | Boucher et al. (2020)     |
| 10  | MIROC6          | $1.4^{\circ} \times 1.4^{\circ}$   | 81                 | Tatebe et al. (2019)      |
| 11  | MIROC-ES2L      | $2.8125^\circ \times 2.79^\circ$   | 40                 | Hajima et al. (2020)      |
| 12  | MPI-ESM1-2-HR   | $1.875^\circ \times 1.865^\circ$   | 47                 | Müller et al. (2018)      |
| 13  | MPI-ESM1-2-LR   | $0.9375^\circ \times 0.935^\circ$  | 85                 | Mauritsen et al. (2019)   |
| 14  | MRI-ESM2-0      | $1.125^\circ \times 1.121^\circ$   | 80                 | Kawai et al. (2019)       |
| 15  | NESM3           | $1.875^\circ \times 1.865^\circ$   | 47                 | Yang et al. (2020)        |
| 16  | NorESM2-LM      | $2.5^{\circ} \times 1.895^{\circ}$ | 32                 | Seland et al. (2020)      |
| 17  | NorESM2-MM      | $1.25^{\circ}\times0.942^{\circ}$  | 32                 | Seland et al. (2020)      |
| 18  | UKESM1-0-LL     | $1.875^\circ\times1.25^\circ$      | 85                 | Sellar et al. (2020)      |

(1980-2014) and SSP585 (2015-2050) scenarios. These additional data are needed to convert the wind from model levels to
heights about model ground levels, e.g. the turbine hub height. Wind speeds above ground level at 50 m, 100 m and 200 m are obtained through vertical logarithmic height interpolation; these heights are relevant for modern turbine sizes.

To reduce the data download volume, the CMIP6 model data was cropped to approximately cover the area  $-10 - 30^{\circ}$ E and  $50 - 70^{\circ}$ N, same as in Hahmann et al. (2022). The 18 models are labeled with numbers in Table 1, together with the corresponding model atmospheric grid spacing and the number of vertical model levels. All data are available on 6-hour basis. Details of these models can be found in the references.

# 2.2 The reanalysis data

The reanalysis data is used to assess the reliability of the CMIP6 wind speeds in describing extreme winds in this region by qualitatively examining the spatial distribution and patterns of  $U_{50}$ .

Three reanalysis data sets are used: (1), the CFSR data (Saha et al., 2010) available at hourly temporal resolution with a grid spacing of about 40 km; (2), the MERRA2 data (Gelaro et al., 2017) available hourly with a grid spacing of  $0.5 \times 0.625^{\circ}$ ; (3),

the ERA5 (Hersbach et al., 2020) available hourly with a grid spacing of about 27 km. We calculate  $U_{50}$  using the his-Period with the three data sets with the SC method.

In the three reanalysis data sets, wind speed diagnostics are available at different heights: 10 m (CFSR), 10 m and 50 m (MERRA2), and at 10 m and 100 m (ERA5). Optimally a single common height for comparison to hub height is necessary. But, the extrapolation of the wind speeds from a height, i.e., 10 m or 50 m, to a typical modern turbine hub height (~100 m) introduces several assumptions (e.g. Hahmann et al., 2022), and thus considerable uncertainty is added when considering the magnitude of the wind speed. Thus, we avoid addressing the absolute value at a given grid point and focus on the spatial patterns of  $U_{50}$  based on the values at 10 m where all three reanalysis data are available.

### 2.3 The spectral correction method

- The IEC standard requires that the 50-year return wind estimate is based on time series equivalent to a temporal resolution of 10-min. Thus, we cannot use the estimation of  $U_{50}$  directly from the 6-hourly CMIP6 data to refer to the IEC standard. The poor temporal sampling means that significant variability is missed compared to time series with a sampling rate every 10-min, which is essential for the estimation of the extreme wind (Larsén and Mann, 2006). When being presented as power spectrum S(f) as a function of frequency f (exemplified in Fig. 1), the original 6-hourly time series shows a fast decrease of energy from
- about f = 1 day<sup>-1</sup> to very little energy at f = 2 day<sup>-1</sup>, and no energy for f > 2 day<sup>-1</sup>. We thus use the spectral correction method to fill the wind variability back to the CMIP6 time series. This method was developed by Larsén et al. (2012), in which it is assumed that that the once-per-year exceedance of the wind speed follows a Poisson process. At a threshold for the rate of once-per-year, such a distribution of the exceedance can be simplified as a Gaussian process. We use this method together with the Annual Maximum Method and the Gumbel distribution to calculate  $U_{50}$ .
- The maximum wind that occurs once a year,  $\overline{U}_{max}$ , is derived as a function of the zero- and second-order spectral moments  $m_0$  and  $m_2$ :

$$\overline{U}_{\max} = \overline{U} + \sqrt{m_0} \sqrt{2 \ln\left(\sqrt{\frac{m_2}{m_0}} T_0\right)},\tag{1}$$

where  $\overline{U}$  is the mean wind speed,  $T_0$  is the basis period of one year and  $m_i$  is the *i*th spectral moment defined by

$$m_i = 2 \int_0^\infty f^i S(f) \, df. \tag{2}$$

Equations 1 and 2 show that  $\overline{U}_{max}$  is significantly affected by the values of S(f) at high frequencies through  $m_2$ . Thus, if we can correct the spectral tail, we can improve the calculation of  $\overline{U}_{max}$ . We use a spectral model to estimate the tail of the distribution:

$$S(f) = a \cdot f^{-5/3},$$
 (3)

which is the mesoscale part of the expression from Larsén et al. (2013) for  $(1 \text{ day})^{-1} 

**Figure 1.** An example of the spectrum of the 6-hourly time series of wind speed at 100 m from the CNRM-CM6-1 model of historical period and the corrected spectrum to equivalent resolution of 10 min. The time series is from the location 56.7309°N, 9.84372°W

tapered-out spectrum in blue for  $f > 0.8 \text{ day}^{-1}$  with Eq. 3 for  $0.8 < f < 72 \text{ day}^{-1}$ , where 72 day<sup>-1</sup> is the Nyquist frequency of a time series with a temporal resolution of 10-min. Then we apply Eq. 1 and 2 with both the original and corrected spectra to obtain  $\overline{U}_{\text{max,orig}}$  and  $\overline{U}_{\text{max,corr}}$ . The ratio of the two is used to correct the annual maximum values from the CMIP6 time series to the values equivalent to 10 min. The magnitude of the change from the spectral correction method depends on how smoothed the original time series is in comparison with what is expected of an observed 10-min time series. For the example shown in Fig. 1,  $U_{50} = 32.5 \text{ m s}^{-1}$  from the raw CMIP6 data and  $U_{50} = 42.8 \text{ m s}^{-1}$  after the spectral correction is applied.

# 3 Results

135

For each model grid point, U<sub>50</sub> are calculated from the reanalysis data using the spectral correction method, and both U<sub>50</sub> and the 95%-percentile (Q95) are calculated from the CMIP6 data for the his-Period and fut-Period. In the following, results of U<sub>50</sub> from the reanalysis data and CMIP6 of his-Period are presented in section 3.1.1 and 3.1.2, where the quality of the CMIP6 ensemble members is discussed. The CC impact on the extreme wind is analyzed in section 3.2.

# 3.1 Extreme wind based on the historic period

### 3.1.1 Reanalysis data

The 50-year wind  $U_{50}$  at 10 m from the CFSR, MERRA2 and ERA5 reanalysis data are shown in Fig. 2 for the his-Period. 145 Over water, the spatial gradient in  $U_{50}$  is similar in the three data sets, with the highest values northwest of the British Islands and lowest in the northern Baltic Sea. Among the three, the CFSR and MERRA2-derived  $U_{50}$  are more similar with respect to the spatial patterns and magnitudes. The ERA5  $U_{50}$  at 10 m is on average systematically smaller. Over land, an obvious