# Peer review of "The Impact of Climate Change on Extreme Winds over Northern Europe According to CMIP6"

_Wind Energy Science, 2022_

## Author Comment (AC1)

**Response to the comments from reviewer 1**

Thanks for your comments on our manuscript. Here is our response to each of your comments. The response is in black color.

**General comments**

The authors analyze the effect of climate change on extreme winds in Europe. The analysis is based on a 18 member subset of the CMIP6 ensemble that had previously been used in a related study by Hahmann et al. The authors compare near future (up to 2050) to historical extreme winds, using a temporal downscaling method to compute the 10 min wind speed extreme with a recurrence time of 50 years from 6 hourly climate model outputs. The authors report an increase in the North Sea and parts of the Baltic and reductions over the Scandinavian Peninsula, as well as generally low signal-to-noise ratios. The topic appears highly relevant both in terms of wear & tear and turbine selection. However, I have substantial concerns about the viability of the chosen approach. In addition to the methodological concerns, I find that the paper is very difficult to follow because new methods are introduced on the fly, often in very imprecise terms, making it difficult to understand what exactly happens. In its current form, I doubt that anyone but the authors would be able to reproduce the results. Unless the paper is substantially improved along the issues outlined below, I suggest to reject it.

The authors get an impression that the reviewer might be working on relevant but different subjects, as it seems that our communication to different research groups needs to be improved. For instance, the authors are not familiar with the phrases "wear & tear" that the reviewer wrote – does the reviewer mean "veer and shear" or fatigue and damage?

We will at least in this reply make suggestions for improving our presentation of the study, not only to the wind energy community.

First of all, the paper is about "extreme wind", which is a key design parameter for deciding what kind of turbines to be used for a particular location. We wonder if the reviewer was referring to the spectral correction method for calculating the extreme wind, when he/she said "new methods are introduced on the fly" (related to the reviewer's first specific question). For the spectral correction method, it was developed by the lead author and published in 2012, and since then it has been cited and used not only in researches but also implemented by the industry. We apologize that the reviewer got the impression that it is a "new method" and will elaborate on this when replying the first specific question below.

**Specific Comments**

Below, I provide a list of concerns (roughly decreasing in importance):

1. The approach heavily and non-linearly relies on the spectrum in the sub-daily part. However, the chosen input data only has coarse temporal resolution of 6h. The authors fill the high frequency part of the spectrum using a simple equation (Eq. 3) with one free parameter. There is no evidence or supporting analysis that would justify this approach in the context of using global climate model output. I think there is a risk that a large part of the results is an artifact of this methodological choice. One way to provide evidence that this approach is solid would be to test your spectral tail correction with the real data of those models that provide higher output frequencies (or by artificially reducing CMIP6 output frequency and checking whether your approach reproduces the real spectrum). Moreover, regional climate model simulations would be an obvious alternative since they provide higher resolution in space and time.

   The authors understand the concern of the reviewer and will in the revised version provide more, clear and direct information on the spectral correction method, which has been validated with measurements from several continents from earlier studies, as documented in, e.g., the following literature (which are already included in the paper):

   Larsén, X. G., Ott, S., Badger, J., Hahmann, A. H., and Mann, J.: Recipes for correcting the impact of effective mesoscale resolution on the estimation of extreme winds, J. Appl. Meteorol. Climat., 51, 521–533, https://doi.org/10.1175/JAMC-D-11-090.1, 2012.

   Larsén, X. G. and Kruger, A.: Application of the spectral correction method to reanalysis data in South Africa, J. Wind Eng. Ind. Aerodyn., 133, 110–122, 2014.

   B. O. Hansen, X. G. Larsén, M. Kelly, O. S. Rathmann, J. Berg, A. Bechmann, A. M. Sempreviva, and H. E. Jørgensen. Extreme wind calculation applying spectral correction method - test and validation. Technical report, Wind Energy Department, Technical University of Denmark, DTU Wind Energy E–0098, 2016

   Larsén, X. G., Davis, N., Hannesdottir, A., Kelly, M., Svenningsen, L., Slot, R., Imberger, M., Olsen, B., and Floors, R.: The Global Atlas for Siting Parameters (GASP) project: extreme wind, turbulence and turbine classes, Wind Energy, Volume25, Issue11, Pages 1841-1859, https://doi.org/10.1002/we.2771, 2022.

2. The authors do not bias correct the climate models and they do not provide any explanation why not.

   We agree that we owe the readers an explanation why we did not provide a bias correction of the climate models.

   We will in the revised version emphasize that the purpose of this study is to assess the impact of climate change on the extreme wind from the collection of CMIP6 models. We will also explain that we choose not to bias correct the climate models because:

   (1) More assumptions need to be introduced when bias correcting a specific model output, which also dependent on what reference data to be used, e.g., reanalysis data or measurements

from limited stations. A lot of the effort needs to go to the reliability of the reanalysis data across the domain and the representativeness of the measurements for making such a bias correction over such an area.

(2) It is another assumption one has to make regarding whether the bias in the historic period can be applied for the future scenarios.

(3) If the assumption in (2) is justified and the bias is consistent in both the historic and the future scenarios, then why do we need to correct them, since we are only interested in *the change*?

3. The authors do not thorougly discuss ensemble agreement, do not quantify significance, and do not compare to background climate variability. That is, they fail to address essential elements of any climate change impact study.

The authors agree that major effort has been put on topics driven by wind energy application and less on the climate change. In the revised version, we will add more analysis on the ensemble agreement and quantify the significance.

4. The method is insufficiently documented. Examples:

- Authors state that wind speeds above ground are computed using the logarithmic height profile but details are unclear. Do they use surface roughness from the individual models to do so? Do they include changes in surface roughness due to land use change?

  The 50, 100 and 200 m wind speed is derived as explained in *Hahmann et al. Current and future wind energy resources in the North Sea according to CMIP6. Wind Energy Science, 7, 2373–2391, 2022.* Using an integration of the hypsometric equation, the height of the CMIP6 model sigma levels are determined from the temperature, specific humidity and surface pressure. Knowing the height of each model level, the horizontal wind speed, can then be interpolated to any height above the model terrain using the natural logarithm of the height. Since the model wind speed is interpolated directly from the model wind speed and all models have at least a level below 100 m, no extrapolation or assumption is made regarding the surface roughness is done. We will in the revised version make this part clear.

- Authors do not explain how the value of a in Eq. 3 is derived although it is central to the analysis. Is that a fit? Looking at Fig.1, your approach matches the data fairly badly.

  We agree that the method was not clearly explained. We will provide a revised text to explain this method more straightforwardly. To this specific question: no, it is not a fit. The purpose of this figure is NOT to show the agreement of the two curves, but to show the missing information in the spectrum obtained from the modeled time series (blue), compares to the red one based on measurements. We will, in the revised version, explain that the value of "a" can be obtained from the modeled time series.

- You replace the spectrum for frequencies higher than 0.8 per day. Why? Your data goes up to 4 per day.

  These two numbers are two separate concepts. The model data goes to 4 per day, and that is the problem —- our method wants to correct it to 144 per day to match the IEC standard, where the corresponding value needs to be of equivalent temporal resolution

of 10 min. 0.8 per day is where we replace the "bad" spectrum from the model data (the blue), with the "good" spectrum from what measurements suggest (the red). This is not a usual climate change concept but a very important and standard wind energy concept. We will try to bridge the gap in the new version of the paper.

- You mention Q95 for the first time in your results. How do you define it? On the raw data? Or using your correction?

  Good point. We will, in the revised version, add the definition of Q95, which is based on the raw data. It is not a design parameter, and therefore does not need to go through the correction. The calculation from the raw data is sufficient for detecting the change.

- Out of the four criteria that you define, the first two appear too broad. The range in (a) is huge and stronger winds over oceans as compared to land (b) are also fairly trivial. As you can see in your Table 2, all models score on criteria 1 & 2.

  Indeed, one can see this as unnecessary to include criteria 1 & 2, or as a quality check for all the models.

- Annual/Periodic Maximum Method and Peak over Threshold Method first mentioned in Results and not introduced

  We will add these definitions to the appendix in the revised version, not to take the focus away from the main subject: the impact of climate change on the extreme wind.

- Why do you compare the CMIP6 data to the FINO obs while not comparing the re-analyses to obs?

  The reason is that only through the CMIP6 data can we assess the climate change impact. However, the authors can also see that including the comparison with the reanalysis is useful. We will include it in the revised version.

- Deep into the Results, you mention that you regrid the CMIP6 data to the grid of model 5 using nearest-neighbor. First, this info belongs into the Methods. Second, please provide justification for these choices. What is the rationale to choose a grid that sits in the middle of the range? Why not choose the grid of a different model, normally the one with the coarsest resolution? Also, nearest neighbor interpolation creates artifacts near the coast which can be really essentially if water masses are small compared to model resolution like in the Baltic Sea. Please justify.

  Sure, this part can be placed in the Methods section. The reason for using the grid of model 5 (which is also the same as 6 and 10) is that models 5, 6, 10 got the highest scores — we would rather keep the information in these models and not weaken them. In addition to models 5, 6 and 10, models 1, 9 and 18 also got high scores, but the spatial resolution is coarser. The reviewer mentioned it that "normally the one with the coarsest resolution" is chosen. The authors would argue that making such a choice would filter out much information from the higher resolution model data, while if we choose the highest resolution, we can reserve as much information as possible, without making unnecessary changes. We are aware that we do not gain additional information from the coarser models through this technique.

- You introduce r on the fly and from what I understand it is the relative change in the corrected U50 in the ensemble mean. You then call r greater than 5% a "significant increase". This wording is confusing because it carries the meaning of a statistical significance test which you do not perform (even though you probably should give the large spread between models)

We will introduce $r$ more formerly in the revised version. Note that $r$ is calculated for each model and each grid point. The spread of $r$ is thus from the 18 model members. Note that for each grid point and each model, there is only one value of U50, which is calculated using a sample collection and the Gumbel distribution; so, unlike the mean wind speed, there is no inter-annual variability in U50. Each model suggests a value of $r$ and Fig. 7 shows the suggestions from majority of the model members. This analysis may be a bit different from the standard climate data analysis; however, it would be helpful if the reviewer provides an argument why this is not good for our study (impact of CC on extreme wind) from a mathematical point of view.

5. What is the role of spatial aggregation? CMIP6 model resolution is quite coarse (grid box size around 10 000 km$^2$), and you investigate extremes, don't you need to disaggregate in space as well?

This issue was raised in Introduction; we acknowledge that this would be the most ideal scenario. However, it requires a major computational efforts to downscale the climate model spatially to a satisfactory resolution, e.g. 3 km. This is also the reason why such data are not available yet. While this may take several years until such a study be carried out, we aim at bringing the latest available data to the wind energy community as soon as possible. Here we bring our expertise on the wind energy side and try to improve one thing at a time, while pursuing opportunities (resource and data) for further improvement (spatial downscaling).

6. Attribution: Many of your results are perfectly compatible with climate variability being the cause for the change instead of climate change. For example, Fig. 4 shows that roughly half of the models show weak increase while the other half shows weak decreases. You don't provide any evidence in terms of processes or comparison to background variability to justify the conclusion that your results actually represent climate change. The same is true when looking at Table 3, in particular for those models that score highly (i.e. Group II). Changes in the mean are essentially zero and the standard deviation is 1 to 2 orders of magnitude larger than the change.

We wonder if we understood it correctly what the reviewer means by "background variability". If it is about the large scale variation, e.g., inter-annual variability, then we would agree that it is easy to be addressed if we are studying the mean wind characteristics. As mentioned above, from 30-year data, we can only obtain ONE value of U50, plus uncertainties associated with the fit of a Gumbel distribution.

We will, in the revised version, mention that one needs to read the statistics in Table 3 from different rows jointly. Ideally, if we plot all the distribution of discrepancies from comparison for each model and each grid point, it will be easier to see what differences are systematic and what are random. That will take too much space. Table 3 aims to provide a simplified version of the distribution.

"mean +/- sigma": for instance -0.23ms$^{-1}$ +/- 0.51ms$^{-1}$: Does the sigma need to be smaller than the mean to suggest a systematic negative trend in the whole dataset? Not necessarily. Of course, such comparison can be used to argue the level of confidence in the negative trend.

The simultaneous values of "min, median, max" aim at providing a rough distribution of the whole data sets. Following the above example, the corresponding values are "-1.69, -0.20, 1.08". With available information so far, one can draw a distribution where the data suggest a systematic negative trend.

For that, the phrases in the text reads: "The overall trend over the whole domain suggests slightly weaker extreme wind in the fut-Period averaged over all CMIP6 models." Then in Table 3, the information goes into further details of the spatial distribution, where the discussion is around "the largest chance" or "largest possibility", and the final conclusion based on these numbers is "The pattern elsewhere is unclear".

7. In your Discussion you write: "There are systematic and consistent patterns for increased and decreased extreme winds that can be identified in certain regions, for both U50 and Q95, even though we are using different groups of data from the 18 CMIP6 models." Could you please point to exact figure and/our table to provides evidence for this claim? From what I see, this conclusion is incompatible with your results (see e.g., my comment 6 above).

Hope the reply to comment 6 is useful. Otherwise, we are referring to Fig 5, 6 and 7.

8. Similarly, you then write: "over the entire study domain, an overall decrease in U50 and Q95; the largest model group (about 40%) suggests no considerable change, 20% of models suggests significant increase and a slightly smaller number of models suggests significant decrease". How can you flag an overall decrease in U50 and Q95 as one of your main results if the majority of models (40%) suggests no change and the others do not agree on the sign of change? This sentence contradicts itself.

The authors do not fully follow the comment: "How can you flag an overall decrease in U50 and Q95 as one of your main results if the majority of models (40%) suggests no change and the others do not agree on the sign of change?"

First of all, 40% is for the largest model group, but it is not a majority.

Secondly, one needs to see how the other groups show different signals, and eventually these contradictory suggestions point to an overall increase or decrease. It is thus possible, and also proven by the numbers in Table 3, that we have an overall decrease — remember we have 5 situations (significant increase, medium increase, no change, medium decrease and significant decrease).

But we acknowledge the current text needs improvement and we will work on that in the new version.

9. You say that the results do "not show any dependence on the spatial resolution of the CMIP6 data". This is a very strong statement. What part of your analysis backs this statement up?

Good point. The sentence needs to be re-written. What is meant is the agreement between the CMIP6 data and reanalysis and measurements do not suggest that higher resolution guarantees better agreement.

10. Code availability only points to retrieval of CMIP6 winds developed by Andrea Hahmann for a separate paper. I can't see any code for this analysis.

We will in the revised version provide codes for calculating the extreme wind and corresponding analysis.

11. The paper is heavily self-referential. I count 15 reference to own work.

The authors will, of course, consider to remove some, if the reviewer can point out those that should not be included. We will also be happy to accept suggestions of other relevant references.

12. CMIP6 subselection: I believe to remember that the choice in Hahmann et al. was also motivated by the need for air density to compute wind energy density. This criterion would not seem to matter here so why not add more models?

   In this study, air density is not needed. However, to interpolate the model wind speed on the original sigma level to the desired heights, we use the temperature and specific humidity to calculate the virtual temperature and the height of each model level. These fields are not available for all models and SSP scenarios at the 6 hour time interval.

**Minor comments**

- Review of previous publications (roughly lines 23 - 37) mostly lists publications. It doesn't summarize and/or contextualize the reported results.

   As responded to comment 1, more details will be provided in the new version.

- You are citing an industry study from 2011 as evidence for "damage to society from extreme winds is rising every year". This study is 12 years old. How can it provide evidence that damages rise every year?

   Good point. We will update this with a new reference: Weather-related disasters increase over past 50 years, causing more damage but fewer deaths, by World Meteorological Organization
   (https://public.wmo.int/en/media/press-release/weather-related-disasters-increase-over-past-50-years-causing-more-damage-fewer)

- I find your Group names difficult to memorize and I would also suggest to order them differently. Currently Group I is least strict (all models), Group II is most strict and Group III is somewhere in between.

   Suggestion taken. We will reformulate the names in the revised version.

- "based on infrastructure known from the" − > Replace infrastructure with a more appropriate word like data or so

   Suggestion taken.

- "Exemplified for the time series in Fig. 1" − > Fig. 1 shows a spectrum, not a time series.

   Suggestion taken.

Best regards,
The authors

---

## Author Comment (AC2)

**Response to the comments from reviewer 2**

Thanks for your comments on our manuscript. Here is our response to each of your comments. The response is in black color.

**General Comments**

This study investigated future changes in extreme winds over Northern Europe by analyzing CMIP6 model results. The authors used a method called "spectral correction" to obtain wind speed data at 10-minute interval using 6-hourly model outputs, and then calculated the 50-year wind. By comparing historical simulations and future projections, they discussed possible changes in extreme winds in the region. This research covers an important topic, and it has some merits. However, I have a few major concerns about the methodology and the statistical significance of the results. I recommend a major revision and have listed my specific comments below.

**Specific Comments**

1. Influences from natural climate variability. The authors compared the CMIP6 historical simulations with reanalysis data over the period 1980–2009 to assess the model performance. However, the observational results during 30-year period are likely affected by both anthropogenic forcing and decadal to inter-decadal natural climate variability. But phases of natural climate variabilities are randomly distributed in the models and therefore not synchronized with observations. In particular, given the relatively small sample size (6 models selected), comparisons between the model and observational results may be strongly affected by low-frequency natural climate variabilities (such as Pacific Decadal Oscillation, Atlantic Multidecadal Oscillation). Similarly, comparing future projections during 2020-2049 with historical simulations may not necessarily isolate the climate change effect, either.

   We will break the above comment into the following questions:
   1) what is the impact of natural climate variability on the extreme wind? 2) how is the low-frequency natural climate variability relevant for comparing the 6 selected models and observational results? 3) Can 2020-2049 vs historical simulations provide information on climate change effect?

   Indeed, the natural climate variability was not examined in the calculation of the extreme wind. Probably it needs to be pointed out that U50 is obtained from a sample of annual maxima with a Gumbel distribution fit. This means that with 30-year data, we obtain ONE value of U50. Uncertainty related to the "low-frequency" variability will be expected in connection with such a calculation (with 30-year data) when the occurrence period of this "low-frequency" event is longer than 30 years. Indeed, if such effect is present, it is difficult to separate if it is the multidecadal variability or climate change.

It is such an interesting subject and we will include the discussion in the Discussion session and at the same time, we will look into and analyze the model data to see if such variabilities are present and if so, what is the effect of using 30-year data instead of a longer period.

On the other hand, the focus of this study is to investigate the change in U50 from historic period to the future period, and therefore it is of secondary importance to find out what caused the differences.

We also would like to point it out that when comparing with measurements, we used 18 (not 6) models (e.g., Fig.4). There are different groups of models formed based on some criteria and the purpose was also to see how this grouping affect the results. However, the results seem to be in general consistent. Apparently we need to improve the corresponding text.

2. Since CMIP models still exhibit biases in simulating many aspects of the global climate system, it is important to select the ones that can better simulate the subject of the research. However, the criteria used to evaluate the CMIP6 model performance (L161-168) seem a bit subjective to me. I would suggest to try using spatial correlation coefficient to assess the model-data agreement, and I wonder if how this may affect the conclusions in this study.

It is a relevant question whether, or how, these biases in the different aspects of the global climate system as in the CMIP models affect our conclusions.

The criteria from L161 – 168 are not enough to evaluate the CMIP6 model performance, as this is also reflected by the comparison with measurements from the three FINO stations (Fig. 4). We did not try to find out which model is best; the criteria and the corresponding grouping rather serve as sensitivity test how the conclusions depend on such grouping. We will make this clear in the revised version.

The reviewer suggests to use spatial correlation coefficient to assess the model-data agreement. The authors are not sure how to bring added values to this subject: it is quite difficult to find out which model can best represent the climate change impact regarding the change in the extreme wind – that's the reason we use an ensemble of models and tried different groupings, as also mentioned in the previous paragraph. Another question is how to transfer the agreement for the historic period to the future period, as the subject is on the climate change effect.

3. I find that there is overall lack of statistical significance test in the results. For instance, L204-209 uses relative changes in the wind speed to determine whether the changes are significant. But even if a change is small in magnitude, it still can be significant as long as the signal is large compared to the noise. I would suggest to perform either a student's t test or show model agreement in the figures to better demonstrate what signals are significant, which may provide more useful information.

We will consider using a student's t test and some other tests to add more analysis on the statistical significance in the revised version.

4. L179-190: Here the authors compared their results with FINO masts, but I think more information about the data is needed to help the readers better understand the results, such as locations, observational periods etc.

More information about the FINO masts will be added. We will also compare the CMIP model results to reanalysis as written in the response to reviewer #1.

Best regards,
The authors